# VALID $P$-VALUE FOR DEEP LEARNING-DRIVEN SALIENT REGION

**Daiki Miwa**[*]
Nagoya Institute of Technology
miwa.daiki.mllab.nit@gmail.com

**Vo Nguyen Le Duy**[*]
RIKEN
duy.vo@riken.jp

**Ichiro Takeuchi**[†]
Nagoya University and RIKEN
ichiro.takeuchi@mae.nagoya-u.ac.jp

## ABSTRACT

Various saliency map methods have been proposed to interpret and explain predictions of deep learning models. Saliency maps allow us to interpret which parts of the input signals have a strong influence on the prediction results. However, since a saliency map is obtained by complex computations in deep learning models, it is often difficult to know how reliable the saliency map itself is. In this study, we propose a method to quantify the reliability of a salient region in the form of $p$-values. Our idea is to consider a salient region as a selected hypothesis by the trained deep learning model and employ the selective inference framework. The proposed method can provably control the probability of false positive detections of salient regions. We demonstrate the validity of the proposed method through numerical examples in synthetic and real datasets. Furthermore, we develop a Keras-based framework for conducting the proposed selective inference for a wide class of CNNs without additional implementation cost.

## 1 INTRODUCTION

Deep neural networks (DNNs) have exhibited remarkable predictive performance in numerous practical applications in various domains owing to their ability to automatically discover the representations needed for prediction tasks from the provided data. To ensure that the decision-making process of DNNs is transparent and easy to understand, it is crucial to effectively explain and interpret DNN representations. For example, in image classification tasks, obtaining *salient regions* allows us to explain which parts of the input image strongly influence the classification results.

Several saliency map methods have been proposed to explain and interpret the predictions of DNN models (Ribeiro et al., 2016; Bach et al., 2015; Doshi-Velez & Kim, 2017; Lundberg & Lee, 2017; Zhou et al., 2016; Selvaraju et al., 2017). However, the results obtained from saliency methods are fragile (Kindermans et al., 2017; Ghorbani et al., 2019; Melis & Jaakkola, 2018; Zhang et al., 2020; Dombrowski et al., 2019; Heo et al., 2019). Therefore, it is important to develop a method for quantifying the reliability of DNN-driven salient regions.

Our idea is to interpret salient regions as hypotheses driven by a trained DNN model and employ a statistical hypothesis testing framework. We use the $p$-value as a criterion to quantify the statistical reliability of the DNN-driven hypotheses. Unfortunately, constructing a valid statistical test for DNN-driven salient regions is challenging because of the *selection bias*. In other words, because the trained DNN *selects* the salient region based on the provided data, the post-selection assessment of importance is biased upwards.

To correct the selection bias and compute valid $p$-values for DNN-driven salient regions, we introduce a conditional *selective inference* (SI) approach. The selection bias is corrected by conditional

---

[*]Equal contribution
[†]Corresponding author

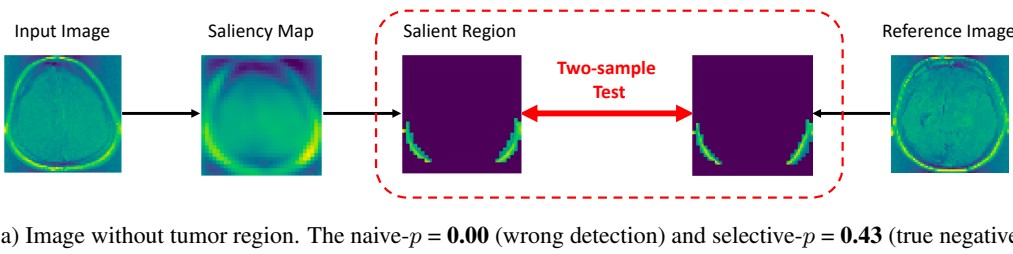

(a) Image without tumor region. The naive-$p$ = **0.00** (wrong detection) and selective-$p$ = **0.43** (true negative)

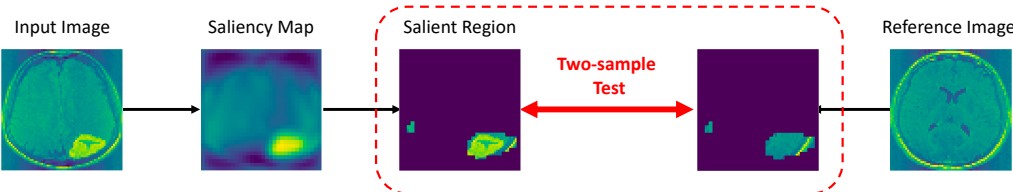

(b) Image with tumor region. The naive-$p$ = **0.00** (true positive) and selective-$p$ = **0.00** (true positive)

Figure 1: Examples of the problem setup and the proposed method on the brain tumor dataset. By applying a saliency method called CAM (Zhou et al., 2016) on a query input image, we obtain the salient region. Our goal is to provide the statistical significance of the salient region in the form of $p$-value by considering two-sample test between the salient region and the corresponding region in the reference image. Note that, since the salient region is selected based on the data, the degree of saliency in the selected region is biased upward. In the upper image where there is no true brain tumor, the naive $p$-value which is obtained without caring about the selection bias is nearly zero, indicating the false positive finding of the salient region. On the other hand, the selective $p$-value which is obtained by the proposed conditional SI approach is $0.43$, indicating that the selected saliency region is not statistically significant. In the lower image where there is a true brain tumor, both the naive $p$-value and the selective $p$-value are very small, which indicate a true positive finding. These results illustrate that naive $p$-value cannot be used to quantify the reliability of DNN-based salient region. In contrast, with the selective $p$-values, we can successfully identify false positive and true positive detections with a desired error rate.

SI in which the test statistic conditional on the event that the hypotheses (salient regions) are selected using the trained DNNs is considered. Our main technical contribution is to develop a method for explicitly deriving the exact (non-asymptotic) conditional sampling distribution of the salient region for a wide class convolutional neural networks (CNNs), which enables us to conduct conditional SI and compute valid $p$-values. Figure 1 presents an example of the problem setup.

**Related works.** In this study, we focus on statistical hypothesis testing for post-hoc analysis, i.e., quantifying the statistical significance of the salient regions identified in a trained DNN model when a test input instance is fed into the model. Several methods have been developed to visualize and understand trained DNNs. Many of these post-hoc approaches (Mahendran & Vedaldi, 2015; Zeiler & Fergus, 2014; Dosovitskiy & Brox, 2016; Simonyan et al., 2013) have focused on developing visualization tools for saliency maps given a trained DNN. Other methods have aimed to identify the discriminative regions in an input image given a trained network (Selvaraju et al., 2017; Fong & Vedaldi, 2017; Zhou et al., 2016; Lundberg & Lee, 2017). However, some recent studies have shown that many of these saliency methods are not stable against a perturbation or adversarial attack on the input data and model (Kindermans et al., 2017; Ghorbani et al., 2019; Melis & Jaakkola, 2018; Zhang et al., 2020; Dombrowski et al., 2019; Heo et al., 2019). To the best of our knowledge, no study to date has succeeded in quantitatively evaluating the reproducibility of DNN-driven salient regions with a rigorous statistical inference framework.

In recent years, conditional SI has emerged as a promising approach for evaluating the statistical reliability of data-driven hypotheses. It has been actively studied for making inferences on the features of linear models selected by various feature selection methods, such as Lasso (Lee et al., 2016). The main concept behind conditional SI is to make inference based on the sampling distribution of the test statistic conditional on a selection event. This approach allows us to derive the exact sampling distribution of the test statistic. After the seminal work of Lee et al. (2016), conditional SI has also

been applied to a wide range of problems (Loftus, 2015; Choi et al., 2017; Tian & Taylor, 2018; Yang et al., 2016; Tibshirani et al., 2016; Fithian et al., 2014; Loftus & Taylor, 2014; Panigrahi et al., 2016; Sugiyama et al., 2021a; Hyun et al., 2021; Duy & Takeuchi, 2021a;b; Sugiyama et al., 2021b; Chen & Bien, 2019; Tsukurimichi et al., 2021; Tanizaki et al., 2020; Duy et al., 2020; 2022).

The most relevant existing work is Duy et al. (2022), where the authors provide a framework to compute valid $p$-values for DNN-based image segmentation results. In Duy et al. (2022), the authors only considered the inference on the output of a DNN in a segmentation task. In this paper, we address a more general problem in which the hypotheses characterized by any internal nodes of the DNN can be considered. This enables us to quantify the statistical significance of salient regions. Furthermore, we introduce a Keras-based implementation framework that enables us to conduct SI for a wide class of CNNs without additional implementation costs. This is in contrast to Duy et al. (2022) in which the selection event must be implemented when the network architecture is changed.

In another direction, Burns et al. (2020) considered the black box model interpretability as a multiple hypothesis testing problem. Their goal was to identify important features by testing the significance of the difference between the prediction and the expected outcome when certain features are replaced with their counterfactuals. However, this approach faces a significant challenge: the number of hypotheses to be considered can be very large (e.g., in the case of an image with $n$ pixels, the number of possible salient regions is $2^n$). Multiple testing correction methods, such as the Bonferroni correction, are highly conservative when the number of hypotheses is large. To address the challenge, they only considered a tractable number of regions selected by a human expert or object detector, which causes selection bias because the candidate regions are selected based on the data.

**Contribution.** Our main contributions are as follows:

• We provide an exact (non-asymptotic) inference method for salient regions obtained by CAM based on the SI concept. We introduce valid $p$-values to statistically quantify the reliability of the DNN-driven salient regions inspired by Duy et al. (2022).

• We propose a novel algorithm and its implementation. Specifically, we propose Keras-based implementation which enables us to conduct conditional SI for a wide class of CNNs without additional implementation costs.

• We conducted experiments on synthetic and real-world datasets, through which we show that our proposed method can control the false positive rate, has good performance in terms of computational efficiency, and provides good results in practical applications. Our code is available at

https://github.com/takeuchi-lab/selective_inference_dnn_salient_region.

## 2 PROBLEM FORMULATION

In this paper, we consider the problem of quantifying the statistical significance of the salient regions identified by a trained DNN model when a test input instance is fed into the model. Consider an $n$-dimensional *query* input vector

$$\boldsymbol{X} = (X_1, ..., X_n)^\top = \boldsymbol{s} + \boldsymbol{\varepsilon}, \quad \boldsymbol{\varepsilon} \sim \mathbb{N}(\boldsymbol{0}, \sigma^2 I_n)$$

and an $n$-dimensional *reference* input vector,

$$\boldsymbol{X}^{\text{ref}} = (X_1^{\text{ref}}, ..., X_n^{\text{ref}})^\top = \boldsymbol{s}^{\text{ref}} + \boldsymbol{\varepsilon}^{\text{ref}}, \quad \boldsymbol{\varepsilon}^{\text{ref}} \sim \mathbb{N}(\boldsymbol{0}, \sigma^2 I_n),$$

where $\boldsymbol{s}, \boldsymbol{s}^{\text{ref}} \in \mathbb{R}^n$ are the signals and $\boldsymbol{\varepsilon}, \boldsymbol{\varepsilon}^{\text{ref}} \in \mathbb{R}^n$ are the noises for query and reference input vectors, respectively. We assume that the signals, $\boldsymbol{s}$ and $\boldsymbol{s}^{\text{ref}}$ are unknown, whereas the distribution of noises $\boldsymbol{\varepsilon}$ and $\boldsymbol{\varepsilon}^{\text{ref}}$ are known (or can be estimated from external independent data) to follow $\mathbb{N}(\boldsymbol{0}, \sigma^2 I_n)$, an $n$-dimensional normal distribution with a mean vector $\boldsymbol{0}$ and covariance matrix $\sigma^2 I_n$, which are mutually independent. In the illustrative example presented in §1, $\boldsymbol{X}$ is a query brain image for a potential patient (we do not know whether she/he has a brain tumor), whereas $\boldsymbol{X}^{\text{ref}}$ is a brain image of a healthy person without brain tumors.

Consider a saliency method for a trained CNN. We denote the saliency method as a function $\mathcal{A} : \mathbb{R}^n \to \mathbb{R}^n$ that takes a query input vector $\boldsymbol{X} \in \mathbb{R}^n$ and returns the saliency map $\mathcal{A}(\boldsymbol{X}) \in \mathbb{R}^n$. We

define a *salient region* $\mathcal{M}_{\boldsymbol{X}}$ for the query input vector $\boldsymbol{X}$ as the set of elements whose saliency map value is greater than a threshold

$$\mathcal{M}_{\boldsymbol{X}} = \{i \in [n] : \mathcal{A}_i(\boldsymbol{X}) \geq \tau\}, \tag{1}$$

where $\tau \in \mathbb{R}$ denotes the given threshold. In this study, we consider CAM (Zhou et al., 2016) as an example of saliency method and threshold-based definition of the salient region. Our method can be applied to other saliency methods and other definitions of salient region.

**Statistical inference.** To quantify the statistical significance of the saliency region $\mathcal{M}_{\boldsymbol{X}}$, we consider a *two-sample test* for the difference between the salient regions of the query input vector $\boldsymbol{X}_{\mathcal{M}_{\boldsymbol{X}}}$ and corresponding region of the reference input vector $\boldsymbol{X}_{\mathcal{M}_{\boldsymbol{X}}}^{\mathrm{ref}}$ where $\boldsymbol{X}_{\mathcal{M}_{\boldsymbol{X}}}$ is a sub-vector of $\boldsymbol{X}$ indexed by $\boldsymbol{X}$. As examples of the two-sample test, we consider the *mean null test*:

$$\mathrm{H}_0 : \frac{1}{|\mathcal{M}_{\boldsymbol{X}}|} \sum_{i \in \mathcal{M}_{\boldsymbol{X}}} s_i = \frac{1}{|\mathcal{M}_{\boldsymbol{X}}|} \sum_{i \in \mathcal{M}_{\boldsymbol{X}}} s_i^{\mathrm{ref}} \quad \text{v.s.} \quad \mathrm{H}_1 : \frac{1}{|\mathcal{M}_{\boldsymbol{X}}|} \sum_{i \in \mathcal{M}_{\boldsymbol{X}}} s_i \neq \frac{1}{|\mathcal{M}_{\boldsymbol{X}}|} \sum_{i \in \mathcal{M}_{\boldsymbol{X}}} s_i^{\mathrm{ref}}, \tag{2}$$

and *global null test*:

$$\mathrm{H}_0 : s_i = s_i^{\mathrm{ref}}, \; \forall i \in \mathcal{M}_{\boldsymbol{X}}, \quad \text{v.s.} \quad \mathrm{H}_1 : s_i \neq s_i^{\mathrm{ref}}, \; \exists i \in \mathcal{M}_{\boldsymbol{X}}. \tag{3}$$

In the mean null test in Eq. (2), we consider a null hypothesis that the average signals in the salient region $\mathcal{M}_{\boldsymbol{X}}$ are the same between $\boldsymbol{X}$ and $\boldsymbol{X}^{\mathrm{ref}}$. In contrast, in the global null test in Eq. (3), we consider a null hypothesis that all elements of the signals in the salient region $\mathcal{M}_{\boldsymbol{X}}$ are the same between $\boldsymbol{X}$ and $\boldsymbol{X}^{\mathrm{ref}}$. The $p$-values for these two-sample tests can be used to quantify the statistical significance of the salient region $\mathcal{M}_{\boldsymbol{X}}$.

**Test-statistic.** For a two-sample test conducted between $\boldsymbol{X}_{\mathcal{M}_{\boldsymbol{X}}}$ and $\boldsymbol{X}_{\mathcal{M}_{\boldsymbol{X}}}^{\mathrm{ref}}$, we consider a class of test statistics called *conditionally linear test-statistic*, which is expressed as

$$T(\boldsymbol{X}, \boldsymbol{X}^{\mathrm{ref}}) = \boldsymbol{\eta}_{\mathcal{M}_{\boldsymbol{X}}}^{\top} \begin{pmatrix} \boldsymbol{X} \\ \boldsymbol{X}^{\mathrm{ref}} \end{pmatrix}, \tag{4}$$

and *conditionally $\chi$ test-statistic*, which is expressed as

$$T(\boldsymbol{X}, \boldsymbol{X}^{\mathrm{ref}}) = \sigma^{-1} \left\| P_{\mathcal{M}_{\boldsymbol{X}}} \begin{pmatrix} \boldsymbol{X} \\ \boldsymbol{X}^{\mathrm{ref}} \end{pmatrix} \right\|, \tag{5}$$

where $\boldsymbol{\eta}_{\mathcal{M}_{\boldsymbol{X}}} \in \mathbb{R}^{2n}$ is a vector and $P_{\mathcal{M}_{\boldsymbol{X}}} \in \mathbb{R}^{2n \times 2n}$ is a projection matrix that depends on $\mathcal{M}_{\boldsymbol{X}}$. The test statistics for the mean null tests and the global null test can be written in the form of Eqs. (4) and (5), respectively. For the mean null test in Eq. (2), we consider the following test-statistic

$$T(\boldsymbol{X}, \boldsymbol{X}^{\mathrm{ref}}) = \boldsymbol{\eta}_{\mathcal{M}_{\boldsymbol{X}}}^{\top} \begin{pmatrix} \boldsymbol{X} \\ \boldsymbol{X}^{\mathrm{ref}} \end{pmatrix} = \frac{1}{|\mathcal{M}_{\boldsymbol{X}}|} \sum_{i \in \mathcal{M}_{\boldsymbol{X}}} X_i - \frac{1}{|\mathcal{M}_{\boldsymbol{X}}|} \sum_{i \in \mathcal{M}_{\boldsymbol{X}}} X_i^{\mathrm{ref}},$$

where $\boldsymbol{\eta}_{\mathcal{M}_{\boldsymbol{X}}} = \frac{1}{|\mathcal{M}_{\boldsymbol{X}}|} \begin{pmatrix} \mathbf{1}_{\mathcal{M}_{\boldsymbol{X}}}^n \\ -\mathbf{1}_{\mathcal{M}_{\boldsymbol{X}}}^n \end{pmatrix} \in \mathbb{R}^{2n}$ with $\mathbf{1}_{\mathcal{C}}^n$ being the $n$-dimensional vector whose elements belongs to the set $\mathcal{C}$ are set to 1, and 0 otherwise. For the global null test in Eq. (3), we consider the following test-statistic

$$T(\boldsymbol{X}, \boldsymbol{X}^{\mathrm{ref}}) = \sigma^{-1} \left\| P_{\mathcal{M}_{\boldsymbol{X}}} \begin{pmatrix} \boldsymbol{X} \\ \boldsymbol{X}^{\mathrm{ref}} \end{pmatrix} \right\| = \sqrt{\sum_{i \in \mathcal{M}_{\boldsymbol{X}}} \left( \frac{X_i - X_i^{\mathrm{ref}}}{\sqrt{2}\sigma} \right)^2},$$

where $P_{\mathcal{M}_{\boldsymbol{X}}} = \frac{1}{2} \begin{pmatrix} \mathrm{diag}(\mathbf{1}_{\mathcal{M}_{\boldsymbol{X}}}^n) & -\mathrm{diag}(\mathbf{1}_{\mathcal{M}_{\boldsymbol{X}}}^n) \\ -\mathrm{diag}(\mathbf{1}_{\mathcal{M}_{\boldsymbol{X}}}^n) & \mathrm{diag}(\mathbf{1}_{\mathcal{M}_{\boldsymbol{X}}}^n) \end{pmatrix}$. To obtain $p$-values for these two-sample tests we need to know the sampling distribution of the test-statistics. Unfortunately, it is challenging to derive the sampling distributions of test-statistics because they depend on the salient region $\mathcal{M}_{\boldsymbol{X}}$, which is obtained through a complicated calculation in the trained CNN.

## 3 COMPUTING VALID $p$-VALUE BY CONDITIONAL SELECTIVE INFERENCE

In this section, we introduce an approach to compute the valid $p$-values for the two-sample tests for the salient region $\mathcal{M}_{\boldsymbol{X}}$ between the query input vector $\boldsymbol{X}$ and the reference input vector $\boldsymbol{X}^{\mathrm{ref}}$ based on the concept of conditional SI (Lee et al., 2016).

### 3.1 CONDITIONAL DISTRIBUTION AND SELECTIVE $p$-VALUE

**Conditional distribution.** The basic idea of conditional SI is to consider the sampling distribution of the test-statistic conditional on a *selection event*. Specifically, we consider the sampling property of the following conditional distribution

$$T(\boldsymbol{X}, \boldsymbol{X}^{\text{ref}}) \mid \{\mathcal{M}_{\boldsymbol{X}} = \mathcal{M}_{\boldsymbol{X}_{\text{obs}}}\}, \tag{6}$$

where $\boldsymbol{X}_{\text{obs}}$ is the observation (realization) of random vector $\boldsymbol{X}$. The condition in Eq.(6) indicates the randomness of $\boldsymbol{X}$ conditional on the event that the same salient region $\mathcal{M}_{\boldsymbol{X}}$ as the observed $\mathcal{M}_{\boldsymbol{X}^{\text{obs}}}$ is obtained. By conditioning on the salient region $\mathcal{M}_{\boldsymbol{X}}$, derivation of the sampling distribution of the conditionally linear and $\chi$ test-statistic $T(\boldsymbol{X}, \boldsymbol{X}^{\text{ref}})$ is reduced to a derivation of the distribution of linear function and quadratic function of $(\boldsymbol{X}, \boldsymbol{X}^{\text{ref}})$, respectively.

**Selective $p$-value.** After considering the conditional sampling distribution in (6), we introduce the following *selective p-value*:

$$p_{\text{selective}} = \mathbb{P}_{\text{H}_0}\Big(\big|T(\boldsymbol{X}, \boldsymbol{X}^{\text{ref}})\big| \geq \big|T(\boldsymbol{X}_{\text{obs}}, \boldsymbol{X}^{\text{ref}}_{\text{obs}})\big| \ \Big| \ \mathcal{M}_{\boldsymbol{X}} = \mathcal{M}_{\boldsymbol{X}_{\text{obs}}}, \mathcal{Q}_{\boldsymbol{X}, \boldsymbol{X}^{\text{ref}}} = \mathcal{Q}_{\text{obs}}\Big), \tag{7}$$

where

$$\mathcal{Q}_{\boldsymbol{X}, \boldsymbol{X}^{\text{ref}}} = \Omega_{\boldsymbol{X}, \boldsymbol{X}^{\text{ref}}}, \quad \mathcal{Q}_{\text{obs}} = \mathcal{Q}_{\boldsymbol{X}_{\text{obs}}, \boldsymbol{X}^{\text{ref}}_{\text{obs}}}$$

with $\Omega_{\boldsymbol{X}, \boldsymbol{X}^{\text{ref}}} = \left(I_{2n} - \frac{\boldsymbol{\eta}_{\mathcal{M}_{\boldsymbol{X}}} \boldsymbol{\eta}_{\mathcal{M}_{\boldsymbol{X}}}^{\top}}{\|\boldsymbol{\eta}_{\mathcal{M}_{\boldsymbol{X}}}\|^2}\right)\left(\begin{smallmatrix}\boldsymbol{X}\\\boldsymbol{X}^{\text{ref}}\end{smallmatrix}\right) \in \mathbb{R}^{2n}$ in the case of mean null test, and

$$\mathcal{Q}_{\boldsymbol{X}, \boldsymbol{X}^{\text{ref}}} = \{\mathcal{V}_{\boldsymbol{X}, \boldsymbol{X}^{\text{ref}}}, \mathcal{U}_{\boldsymbol{X}, \boldsymbol{X}^{\text{ref}}}\}, \quad \mathcal{Q}_{\text{obs}} = \mathcal{Q}_{\boldsymbol{X}_{\text{obs}}, \boldsymbol{X}^{\text{ref}}_{\text{obs}}}$$

with $\mathcal{V}_{\boldsymbol{X}, \boldsymbol{X}^{\text{ref}}} = \sigma P_{\mathcal{M}_{\boldsymbol{X}}}\left(\begin{smallmatrix}\boldsymbol{X}\\\boldsymbol{X}^{\text{ref}}\end{smallmatrix}\right)\Big/\left\|P_{\mathcal{M}_{\boldsymbol{X}}}\left(\begin{smallmatrix}\boldsymbol{X}\\\boldsymbol{X}^{\text{ref}}\end{smallmatrix}\right)\right\| \in \mathbb{R}^{2n}$, $\mathcal{U}_{\boldsymbol{X}, \boldsymbol{X}^{\text{ref}}} = P^{\perp}_{\mathcal{M}_{\boldsymbol{X}}}\left(\begin{smallmatrix}\boldsymbol{X}\\\boldsymbol{X}^{\text{ref}}\end{smallmatrix}\right) \in \mathbb{R}^{2n}$ in the case of global null test. The $\mathcal{Q}_{\boldsymbol{X}, \boldsymbol{X}^{\text{ref}}}$ is the sufficient statistic of the nuisance parameter that needs to be conditioned on in order to tractably conduct the inference [1].

The selective $p$-value in Eq.(7) has the following desired sampling property

$$\mathbb{P}_{\text{H}_0}\Big(p_{\text{selective}} \leq \alpha \mid \mathcal{M}_{\boldsymbol{X}} = \mathcal{M}_{\boldsymbol{X}_{\text{obs}}}\Big) = \alpha, \quad \forall \alpha \in [0, 1]. \tag{8}$$

This means that the selective $p$-values $p_{\text{selective}}$ can be used as a valid statistical significance measure for the salient region $\mathcal{M}_{\boldsymbol{X}}$.

### 3.2 CHARACTERIZATION OF THE CONDITIONAL DATA SPACE

To compute the selective $p$-value in (7), we need to characterize the conditional data space whose characterization is described and introduced in the next section. We define the set of $(\boldsymbol{X}^{\top}\ \boldsymbol{X}^{\text{ref}^{\top}})^{\top} \in \mathbb{R}^{2n}$ that satisfies the conditions in Eq. (7) as

$$\mathcal{D} = \left\{(\boldsymbol{X}^{\top}\ \boldsymbol{X}^{\text{ref}^{\top}})^{\top} \in \mathbb{R}^{2n} \mid \mathcal{M}_{\boldsymbol{X}} = \mathcal{M}_{\boldsymbol{X}_{\text{obs}}} \mathcal{Q}_{\boldsymbol{X}, \boldsymbol{X}^{\text{ref}}} = \mathcal{Q}_{\text{obs}}\right\}. \tag{9}$$

According to the second condition, the data in $\mathcal{D}$ is restricted to a line in $\mathbb{R}^{2n}$ as stated in the following Lemma.

**Lemma 1.** *Let us define* $\boldsymbol{a} = \Omega_{\boldsymbol{X}_{\text{obs}}, \boldsymbol{X}^{\text{ref}}_{\text{obs}}}$ *and* $\boldsymbol{b} = \frac{\boldsymbol{\eta}_{\mathcal{M}_{\boldsymbol{X}}}}{\|\boldsymbol{\eta}_{\mathcal{M}_{\boldsymbol{X}}}\|^2} \in \mathbb{R}^{2n}$ *in the case of mean null test, and* $\boldsymbol{a} = \mathcal{U}_{\boldsymbol{X}_{\text{obs}}, \boldsymbol{X}^{\text{ref}}_{\text{obs}}}$ *and* $\boldsymbol{b} = \mathcal{V}_{\boldsymbol{X}_{\text{obs}}, \boldsymbol{X}^{\text{ref}}_{\text{obs}}}$ *in the case of global null test. The $\mathcal{D}$ in (9) can be rewritten as* $\mathcal{D} = \left\{\left(\boldsymbol{X}^{\top}\ \boldsymbol{X}^{\text{ref}^{\top}}\right)^{\top} = \boldsymbol{a} + \boldsymbol{b}z \mid z \in \mathcal{Z}\right\}$ *by using the scalar parameter $z \in \mathbb{R}$, where*

$$\mathcal{Z} = \{z \in \mathbb{R} \mid \mathcal{M}_{\boldsymbol{a}_{1:n} + \boldsymbol{b}_{1:n}z} = \mathcal{M}_{\boldsymbol{X}_{\text{obs}}}\}. \tag{10}$$

*with $\boldsymbol{x}_{1:n}$ representing a vector of elements $1$ through $n$ of $\boldsymbol{x}$.*

---

[1]This nuisance parameter $\mathcal{Q}_{\boldsymbol{X}, \boldsymbol{X}^{\text{ref}}}$ corresponds to the component $\boldsymbol{z}$ in the seminal conditional SI paper (Lee et al., 2016) (see Sec. 5, Eq. 5.2 and Theorem 5.2) and $\boldsymbol{z}, \boldsymbol{w}$ in (Chen & Bien, 2019)(see Sec. 3, Theorem 3.7). We note that additional conditioning on $\mathcal{Q}_{\boldsymbol{X}, \boldsymbol{X}^{\text{ref}}}$ is a standard approach in the conditional SI literature and is used in almost all conditional SI-related studies. Here, we would like to note that the selective $p$-value depends on $\mathcal{Q}_{\boldsymbol{X}, \boldsymbol{X}^{\text{ref}}}$, but the property in (8) is satisfied without this additional condition because we can marginalize over all values of $\mathcal{Q}_{\boldsymbol{X}, \boldsymbol{X}^{\text{ref}}}$ (see the lower part of the proof of Theorem 5.2 in Lee et al. (2016) and the proof of Theorem 3.7 in Chen & Bien (2019) ).

*Proof.* The proof is deferred to Appendix A.1.

Lemma 1 indicates that we do not need to consider the $2n$-dimensional data space. Instead, we only need to consider the *one-dimensional projected* data space $\mathcal{Z}$ in (10). Now, let us consider a random variable $Z \in \mathbb{R}$ and its observation $Z_{\mathrm{obs}} \in \mathbb{R}$ that satisfies $(\boldsymbol{X}^\top \ \boldsymbol{X}^{\mathrm{ref}^\top})^\top = \boldsymbol{a} + \boldsymbol{b}Z$ and $(\boldsymbol{X}_{\mathrm{obs}}^\top \ \boldsymbol{X}_{\mathrm{obs}}^{\mathrm{ref}^\top})^\top = \boldsymbol{a} + \boldsymbol{b}Z_{\mathrm{obs}}$. The selective $p$-value (7) is rewritten as

$$p_{\mathrm{selective}} = \mathbb{P}_{\mathrm{H}_0}\left(|Z| \geq |Z_{\mathrm{obs}}| \mid Z \in \mathcal{Z}\right). \tag{11}$$

Because $(\boldsymbol{X}^\top \ \boldsymbol{X}^{\mathrm{ref}^\top})^\top \sim N\left(\left(\boldsymbol{s}^\top \ \boldsymbol{s}^{\mathrm{ref}^\top}\right)^\top, \sigma^2 I_{2n}\right)$ due to the independence between $\boldsymbol{X}$ and $\boldsymbol{X}^{\mathrm{ref}}$, the variable $Z \sim \mathbb{N}(0, \sigma^2\|\boldsymbol{\eta}\|^2)$ in the case of mean null test and $Z \sim \chi\left(\mathrm{Trace}(P)\right)$ in the case of global null test under the null hypothesis. Therefore, $Z \mid Z \in \mathcal{Z}$ follows a *truncated* normal distribution and a *truncated* $\chi$ distribution, respectively. Once the truncation region $\mathcal{Z}$ is identified, computation of the selective $p$-value in (11) is straightforward. Therefore, the remaining task is to identify $\mathcal{Z}$.

# 4 PIECEWISE LINEAR NETWORK

The problem of computing selective $p$-values for the selected salient region is cast into the problem of identifying a set of intervals $\mathcal{Z} = \{z \in \mathbb{R} \mid \mathcal{M}_{\boldsymbol{X}(z)} = \mathcal{M}_{\boldsymbol{X}_{\mathrm{obs}}}\}$. Given the complexity of saliency computation in a trained DNN, it seems difficult to obtain $\mathcal{Z}$. In this section, however, we show that this is feasible for a wide class of CNNs.

**Piecewise linear components in CNN.** The key idea is to note that most of basic operations and common activation functions used in a trained CNN can be represented as piecewise linear functions in the following form:

**Definition 1.** *(Piecewise Linear Function) A piecewise linear function $f : \mathbb{R}^n \to \mathbb{R}^m$ is written as:*

$$f(\boldsymbol{X}) = \begin{cases} \Psi_1^f \boldsymbol{X} + \boldsymbol{\psi}_1^f, & \text{if } \boldsymbol{X} \in \mathcal{P}_1^f := \{\boldsymbol{X}' \in \mathbb{R}^n \mid \Delta_1^f \boldsymbol{X}' \leq \boldsymbol{\delta}_1^f\}, \\ \Psi_2^f \boldsymbol{X} + \boldsymbol{\psi}_2^f, & \text{if } \boldsymbol{X} \in \mathcal{P}_2^f := \{\boldsymbol{X}' \in \mathbb{R}^n \mid \Delta_2^f \boldsymbol{X}' \leq \boldsymbol{\delta}_2^f\}, \\ \quad\vdots \\ \Psi_{K(f)}^f \boldsymbol{X} + \boldsymbol{\psi}_{K(f)}^f, & \text{if } \boldsymbol{X} \in \mathcal{P}_{K(f)}^f := \{\boldsymbol{X}' \in \mathbb{R}^n \mid \Delta_{K(f)}^f \boldsymbol{X}' \leq \boldsymbol{\delta}_{K(f)}^f\}, \end{cases}$$

*where $\Psi_k^f$, $\boldsymbol{\psi}_k^f$, $\Delta_k^f$ and $\boldsymbol{\delta}_k^f$ for $k \in [K(f)]$ are certain matrices and vectors with appropriate dimensions, $\mathcal{P}_k^f := \{\boldsymbol{x} \in \mathbb{R}^n \mid \Delta_k^f \boldsymbol{x} \leq \boldsymbol{\delta}_k^f\}$ is a polytope in $\mathbb{R}^n$ for $k \in [K(f)]$, and $K(f)$ is the number of polytopes for the function $f$.*

Examples of piecewise linear components in a trained CNN are shown in Appendix A.2.

**Piecewise Linear Network.**
**Definition 2.** *(Piecewise Linear Network) A network obtained by concatenations and compositions of piecewise linear functions is called* piecewise linear network.

Since the concatenation and the composition of piecewise linear functions is clearly piecewise linear function, the output of any node in the piecewise linear network is written as a piecewise linear function of an input vector $\boldsymbol{X}$. This is also true for the saliency map function $\mathcal{A}_i(\boldsymbol{X}), i \in [n]$ obtained by CAM. Furthermore, as discussed in §4, we can focus on the input vector in the form of $\boldsymbol{X}(z) = \boldsymbol{a}_{1:n} + \boldsymbol{b}_{1:n}z$ which is parametrized by a scalar parameter $z \in \mathbb{R}$. Therefore, the saliency map value for each element is written as a piecewise linear function of the scalar parameter $z$, i.e.,

$$\mathcal{A}_i(\boldsymbol{X}(z)) = \begin{cases} \kappa_1^{\mathcal{A}_i} z + \rho_1^{\mathcal{A}_i}, & \text{if } z \in [L_1^{\mathcal{A}_i}, U_1^{\mathcal{A}_i}], \\ \kappa_2^{\mathcal{A}_i} z + \rho_2^{\mathcal{A}_i}, & \text{if } z \in [L_2^{\mathcal{A}_i}, U_2^{\mathcal{A}_i}], \\ \quad\vdots \\ \kappa_{K(\mathcal{A}_i)}^{\mathcal{A}_i} z + \rho_{K(\mathcal{A}_i)}^f, & \text{if } z \in [L_{K(\mathcal{A}_i)}^{\mathcal{A}_i}, U_{K(\mathcal{A}_i)}^{\mathcal{A}_i}], \end{cases} \tag{12}$$

---

**Algorithm 1** `SI_DNN_Saliency`

---

**Input:** $\boldsymbol{X}^{\mathrm{obs}}, z_{\min}, z_{\max}, \mathcal{T} \leftarrow \emptyset$
1: Obtain $\mathcal{E}_{\mathrm{obs}}$, compute $\boldsymbol{\eta}$ as well as $\boldsymbol{a}$ and $\boldsymbol{b} \leftarrow$ Lemma 1 and initialize: $t = 1, z_t = z_{\min}$
2: **for** $t \leq T$ **do**
3:      Compute $z_{t+1}$ by Auto-Conditioning (see §5)
4:      **if** $\mathcal{E}_{\boldsymbol{X}(z), \boldsymbol{X}^{\mathrm{ref}(z)}} = \mathcal{E}_{\mathrm{obs}}$ in $z \in [z_t, z_{t+1}]$ (by using Eq.(13)) **then**
5:          $\mathcal{T} \leftarrow \mathcal{T} + \{t\}$
6:      **end if**
7:      $t = t + 1$
8: **end for**
9: Identify $\mathcal{Z} \leftarrow \bigcup_{t \in \mathcal{T}} [z_t, z_{t+1}]$
10: $p_{\mathrm{selective}} \leftarrow$ Eq. (11)
**Output:** $p_{\mathrm{selective}}$

---

where $K(\mathcal{A}_i)$ is the number of linear pieces of the piecewise linear function, $\kappa_k^{\mathcal{A}_i}, \rho_k^{\mathcal{A}_i}$ are certain scalar parameters, $[L_k^{\mathcal{A}_i}, U_k^{\mathcal{A}_i}]$ are intervals for $k \in [K(\mathcal{A}_i)]$ (note that a polytope in $\mathbb{R}^n$ is reduced to an interval when it is projected onto one-dimensional space).

This means that, for each piece of the piecewise linear function, we can identify the interval of $z$ such that $\mathcal{A}_i(\boldsymbol{X}(z)) \geq \tau$ as follows [2]

$$z \in \begin{cases} \left[\max\left(L_k^{\mathcal{A}_i}, \left(\tau - \rho_k^{\mathcal{A}_i}\right)/\kappa_k^{\mathcal{A}_i}\right), U_k^{\mathcal{A}_i}\right] & \text{if } \kappa_k^{\mathcal{A}_i} > 0 \\ \left[L_k^{\mathcal{A}_i}, \min\left(U_k^{\mathcal{A}_i}, \left(\tau - \rho_k^{\mathcal{A}_i}\right)/\kappa_k^{\mathcal{A}_i}\right),\right] & \text{if } \kappa_k^{\mathcal{A}_i} < 0 \end{cases} \quad \Rightarrow \quad \mathcal{A}_i(\boldsymbol{X}(z)) \geq \tau. \quad (13)$$

With a slight abuse of notation, let us collectively denote the finite number of intervals on $z \in \mathbb{R}$ that are defined by $L_k^{\mathcal{A}_i}, U_k^{\mathcal{A}_i}, (\tau - \rho_i^{\mathcal{A}_i}/\kappa_k^{\mathcal{A}_i})$ for all $(k, i) \in [K(\mathcal{A}_i)] \times [n]$ as

$$[z_0, z_1], [z_1, z_2], \ldots, [z_{t-1}, z_t], [z_t, z_{t+1}], \ldots, [z_{T-1}, z_T],$$

where $z_{\min} = z_0$ and $z_{\max} = z_T$ are defined such that the probability mass of $z < z_{\min}$ and $z > z_{\max}$ are negligibly small.

**Algorithm.** Algorithm 1 shows how we identify $\mathcal{Z} = \{z \in \mathbb{R} \mid \mathcal{M}_{\boldsymbol{X}(z), \boldsymbol{X}^{\mathrm{ref}(z)}} = \mathcal{M}_{\mathrm{obs}}\}$. We simply check the intervals of $z$ in the order of $[z_0, z_1], [z_1, z_2], ..., [z_{T-1}, z_T]$ to see whether $\mathcal{M}_{\boldsymbol{X}(z)} = \mathcal{M}_{\boldsymbol{X}(z_{\mathrm{obs}})}$ or not in the interval by using Eq.(13). Then, the truncation region $\mathcal{Z}$ in Eq.(10) is given as $\mathcal{Z} = \bigcup_{t \in [T] | \mathcal{E}_{\boldsymbol{X}(z), \boldsymbol{X}^{\mathrm{ref}(z)}} = \mathcal{E}_{\mathrm{obs}} \text{ for } z \in [z_t, z_{t+1}]} [z_t, z_{t+1}]$. In the literature of homotopy method (a.k.a. parametric programming), it is known that the actual computational cost differs significantly from the worst case. A well-known application of the homotopy method in the ML community is the Lasso regularization path, which also has the worst-case computational cost on the exponential order of the number of features, but the actual cost is known to be nearly linear order. Empirically, this also applies to our proposed method.

## 5 IMPLEMENTATION: AUTO-CONDITIONING

The bottleneck of our algorithm is Line 3 in Algorithm 1, where $z_{t+1}$ must be found by considering all relevant piecewise linear components in a complicated trained CNN. The difficulty lies not only in the computational cost but also in the implementation cost. To implement conditional SI in DNNs naively, it is necessary to characterize all operations at each layer of the network as selection events and implement each of them specifically (Duy et al., 2022). To circumvent this difficulty, we introduce a modular implementation scheme called *auto-conditioning*, which is similar to *auto-differentiation* (Baydin et al., 2018) in concept. This enables us to conduct conditional SI for a wide class of CNNs without additional implementation costs.

The basic idea in auto-conditioning is to add a mechanism to compute and maintain the interval $z \in [L_k^f, U_k^f]$ for each piecewise linear component $f$ in the network (e.g., layer API in the Keras

---

[2]For simplicity, we omit the description for the case of $\kappa_k^{\mathcal{A}_i} = 0$. In this case, if $\rho_k^{\mathcal{A}_i} \geq \tau$, then $z \in [L_k^{\mathcal{A}_i}, U_k^{\mathcal{A}_i}] \Rightarrow i \in \mathcal{M}_{\boldsymbol{X}(z)}$.

framework). This enables us to automatically compute the interval $[L_k^f, U_k^f]$ of a piecewise linear function $f$ when it is obtained as concatenation and/or composition of multiple piecewise linear components. If $f$ is obtained by concatenating two piecewise linear functions $f_1$ and $f_2$, we can easily obtain $[L_k^f, U_k^f] = [L_{k_1}^{f_1}, U_{k_1}^{f_1}] \cap [L_{k_2}^{f_2}, U_{k_2}^{f_2}]$. However, if $f$ is obtained as a composition of two piecewise linear functions $f_1$ and $f_2$, the calculation of the interval is given by the following lemma.

**Lemma 2.** *Consider the composition of two piecewise linear functions $f(\boldsymbol{X}(z)) = (f_2 \circ f_1)(\boldsymbol{X}(z))$. Given a real value of $z$, the interval $[L_k^{f_2}, U_k^{f_2}]$ in the input domain of $f_2$ can be computed as*

$$L_{k_2}^{f_2} = \max_{j:(\Delta_{k_2}^{f_2}\boldsymbol{\gamma}^{f_1})_j < 0} \frac{(\boldsymbol{\delta}_{k_2}^{f_2})_j - (\Delta_{k_2}^{f_2}\boldsymbol{\beta}^{f_1})_j}{(\Delta_{k_2}^{f_2}\boldsymbol{\gamma}^{f_1})_j}, \qquad U_{k_2}^{f_2} = \min_{j:(\Delta_{k_2}^{f_2}\boldsymbol{\gamma}^{f_1})_j > 0} \frac{(\boldsymbol{\delta}_{k_2}^{f_2})_j - (\Delta_{k_2}^{f_2}\boldsymbol{\beta}^{f_1})_j}{(\Delta_{k_2}^{f_2}\boldsymbol{\gamma}^{f_1})_j},$$

*where $\boldsymbol{\beta}^{f_1} + \boldsymbol{\gamma}^{f_1}z$ is the output of $f_1$ (i.e., the input of $f_2$). Moreover, $\Delta_{k_2}^{f_2}$ and $\boldsymbol{\delta}_{k_2}^{f_2}$ are obtained by verifying the value of $\boldsymbol{\beta}^{f_1} + \boldsymbol{\gamma}^{f_1}z$. Then, the interval of the composite function is obtained as follows: $[L_k^f, U_k^f] = [L_{k_1}^{f_1}, U_{k_1}^{f_1}] \cap [L_{k_2}^{f_2}, U_{k_2}^{f_2}]$*

The proof is provided in Appendix A.3. Here, the variables $\boldsymbol{\beta}^{f_k}$ and $\boldsymbol{\gamma}^{f_k}$ can be recursively computed through layers as

$$\boldsymbol{\beta}^{f_{k+1}} = \Psi_k^{f_k}\boldsymbol{\beta}^{f_k} + \boldsymbol{\psi}_k^{f_k} \quad \text{and} \quad \boldsymbol{\gamma}^{f_{k+1}} = \Psi_k^{f_k}\boldsymbol{\gamma}^{f_k}.$$

Lemma 2 indicates that the intervals in which $\boldsymbol{X}(z)$ falls in can be *forwardly propagated* through these layers. This means that the lower bound $L_k^{\mathcal{A}_i}$ and upper bound $U_k^{\mathcal{A}_i}$ of the current piece in the piecewise linear function in Eq. (12) can be automatically computed by *forward propagation* of the intervals of the relevant piecewise linear components.

## 6 EXPERIMENT

We only highlight the main results. More details (methods for comparison, network structure, etc.) can be found in the Appendix A.4.

**Experimental setup.** We compared our proposed method with the naive method, over-conditioning (OC) method, and Bonferroni correction. To investigate the false positive rate (FPR), we considered 1000 null images $\boldsymbol{X} = (X_1, ..., X_n)$ and 1000 reference images $\boldsymbol{X}^{\text{ref}} = (X_1^{\text{ref}}, ..., x_n^{\text{ref}})$, where $\boldsymbol{s} = \boldsymbol{s}^{\text{ref}} = \boldsymbol{0}$ and $\boldsymbol{\varepsilon}, \boldsymbol{\varepsilon}^{\text{ref}} \sim \mathbb{N}(\boldsymbol{0}, I_n)$, for each $n \in \{64, 256, 1024, 4096\}$. To investigate the true positive rate (TPR), we set $n = 256$ and generated 1,000 images, in which $s_i = \Delta$ for any $i \in \mathcal{S}$, where $\mathcal{S}$ is the "true" salient region whose location is randomly determined, and $s_i = 0$ for any $i \notin \mathcal{S}$. We set $\Delta \in \{1, 2, 3, 4\}$. Reference images were generated in the same way as in the case of FPR. In all experiments, we set $\tau = 0$ in the mean null test and $\tau = 5$ in the global null test. We set the significance level $\alpha = 0.05$. We used CAM as the saliency method in all experiments.

**Numerical results.** The results of FPR control properties are presented in Fig. 2. The proposed method, OC, and Bonferroni successfully controlled the FPR in both the mean and global null test cases, whereas the naive method could not. Because naive method failed to control the FPR, we no longer considered its TPR. The results of the TPR comparison are shown in Fig. 3. The proposed method has the highest TPR in all cases. The Bonferroni method has the lowest TPR because it is conservative owing to considering the number of all possible hypotheses. The OC method also has a low TPR because it considers several extra conditions, which cause the loss of TPR.

**Real data experiments.** We examined the brain image dataset extracted from the dataset used in Buda et al. (2019), which included 939 and 941 images with and without tumors, respectively. The results of the mean null test are presented in Figs. 4 and 5. The results of the global null test are presented in Figs. 6 and 7. The naive $p$-value remains small even when the image has no tumor region, which indicates that naive $p$-values cannot be used to quantify the reliability of DNN-based salient regions. The proposed method successfully identified false and true positive detections.

## 7 CONCLUSION

In this study, we proposed a novel method to conduct statistical inference on the significance of DNN-driven salient regions based on the concept of conditional SI. We provided a novel algorithm

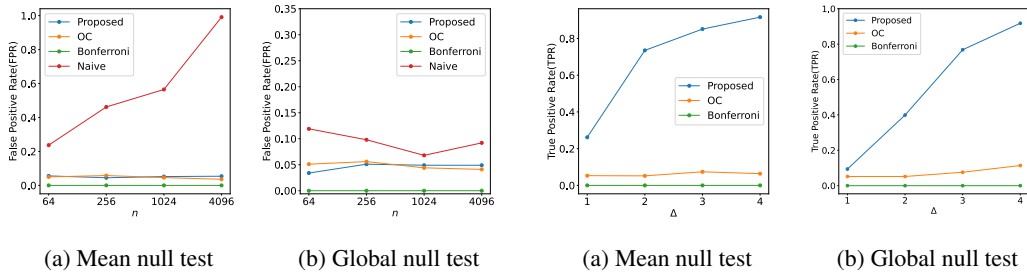

(a) Mean null test      (b) Global null test      (a) Mean null test      (b) Global null test

Figure 2: False Positive Rate (FPR) comparison.      Figure 3: True Positive Rate (FPR) comparison.



Figure 4: Mean null test for image without tumor ($p_{\text{naive}} = \mathbf{0.00}$, $p_{\text{selective}} = \mathbf{0.78}$).

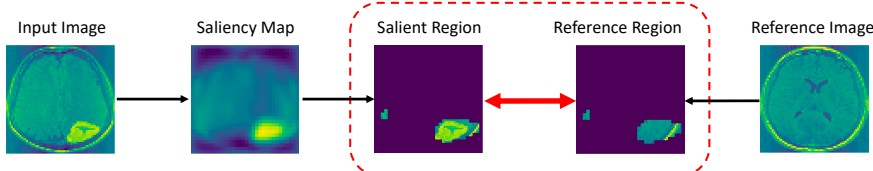

Figure 5: Mean null test for image with a tumor ($p_{\text{naive}} = \mathbf{0.00}$, $p_{\text{selective}} = \mathbf{1.92 \times 10^{-4}}$).



Figure 6: Global null test for image without tumor ($p_{\text{naive}} = \mathbf{0.03}$, $p_{\text{selective}} = \mathbf{0.46}$)



Figure 7: Global null test for image with a tumor ($p_{\text{naive}} = \mathbf{0.00}$, $p_{\text{selective}} = \mathbf{1.51 \times 10^{-3}}$).

for efficiently and flexibly conducting conditional SI for salient regions. We conducted experiments on both synthetic and real-world datasets to demonstrate the performance of the proposed method. In current setting, we have not considered the situations where there is a misalignment between the input image and the reference image. A potential future improvement could be additionally performing a step to automatically find an appropriate region in the reference image before conducting a statistical test. If the matching operations can be represented as a set of linear inequalities, they can be easily incorporated to the proposed method. If the matching operations can be represented as a set of linear inequalities, they can be easily incorporated to the proposed method.

ACKNOWLEDGEMENTS

This work was partially supported by MEXT KAKENHI (20H00601), JST CREST (JPMJCR21D3), JST Moonshot R&D (JPMJMS2033-05), JST AIP Acceleration Research (JPMJCR21U2), NEDO (JPNP18002, JPNP20006), and RIKEN Center for Advanced Intelligence Project.

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
