# OpenReview forum: "Valid P-Value for Deep Learning-driven Salient Region"
_ICLR.cc/2023/Conference — ICLR 2023 poster_

### Official Review · Reviewer_XKd9 · 2022-10-24

**Confidence:** 3
**Correctness:** 3
**Technical Novelty And Significance:** 2
**Empirical Novelty And Significance:** 2
**Recommendation:** 5

**Clarity, Quality, Novelty And Reproducibility:**

The novelty of the paper is limited due to its similarity to the aforementioned NeurIPS paper.

**Strength And Weaknesses:**

Strength:

1. The writing flow of the paper is very nice, including comprehensive background and detailed related work. The motivation is clearly stated.
2. The theoretical results are well presented. The assumptions made for generating such results are clearly described and contextualized.
3. Experimental results consider several reasonable baselines and demonstrate the improvement produced by the method.

Weaknesses:

1. The work relies largely on the previous work to prove an exact inference method for the salient regions. Specifically, relying on similar assumptions and derivation of [1], the paper extends image segmentation to saliency maps. The mathematical formulas, the two-sample test, the conditional distribution, the selective p-value, and the piecewise linear network are all similar to [1]. It looks like an application based on [1]. Thus, it seems that the algorithm itself is not very novel.

2. Is it possible to provide some experimental evidence for the argument that the proposed selective inference works well for a wide class of CNNs without additional implementation cost? How does the method integrate with various CNNs? For this issue, the authors may report the performance of different CNNs. In addition, the proposed method seems computationally expensive. How efficient is the algorithm compared with the baseline methods?

[1] Vo Nguyen Le Duy, Shogo Iwazaki,and Ichiro Takeuchi. Quantifying statistical significance of neural network-based image segmentation by selective inference. NeurIPS, 2022

**Summary Of The Paper:**

This paper focuses on saliency map of class activation. An inference method is proposed to quantify the reliability of a saliency region in the form of p-values based on the concept of SI. The experimental results show the improved performance.

**Summary Of The Review:**

Considering the novelty and the rigorousness of the paper, I hold a slightly negative stance.

---

> ### Author Response · Authors · 2022-11-13
> **Our Responses to Reviewer XKd9**
>
> We thank the reviewer for your feedback.
>
>
> > The work relies largely on the previous work to prove an exact inference method for the salient regions. Specifically, relying on similar assumptions and derivation of Duy et al. (2022), the paper extends image segmentation to saliency maps. The mathematical formulas, the two-sample test, the conditional distribution, the selective p-value, and the piecewise linear network are all similar to Duy et al. (2022). It looks like an application based on Duy et al. (2022). Thus, it seems that the algorithm itself is not very novel.
>
> We would like to clarify the following two novel points:
>
> - We propose a new problem setup, i.e., testing the difference between the salient region in the input image and the corresponding region in the reference image whereas Duy et al. (2022) consider testing the difference between the object and background within the input image. Conceptually, our method is more general in the sense that we can handle the hypotheses characterized by any layer of the network whereas Duy et al. (2022) only handle the hypotheses driven at the output layer (segmentation task). This enables us to quantify the statistical significance of salience regions.
>
> - Another substantial contribution compared to Duy et al. (2022) is the Auto-Conditioning part.  Auto-Conditioning is a trick to ease implementation, not a way to reduce computational cost. Therefore, we have not provided the analysis of computational complexity. Conventional selective inference (including Duy et al. (2022)) requires explicit coding of selection events, so even small changes to the network structure require reimplementation. Our auto-conditioning method takes advantage of the fact that selection events can be computed by forward propagation in each layer. Since selection events are implemented in the layer API of Keras, it is possible to automatically compute selection event. More specifically, the conditioning part of Equation. (9) can be automatically computed by simply adding/removing layers, i.e., just a few line modification is sufficient to conduct selective inference for different network structures (the reviewer can simply add/remove layers of the network in the 'demo.ipynb' file provided in the supplements and the code can be easily run). This auto-conditioning enables selective inference for hypotheses characterized at any layer in the network, which allows us to consider quantification of the statistical significance of saliency regions.
>
>
> > Is it possible to provide some experimental evidence for the argument that the proposed selective inference works well for a wide class of CNNs without additional implementation cost? How does the method integrate with various CNNs? For this issue, the authors may report the performance of different CNNs. In addition, the proposed method seems computationally expensive. How efficient is the algorithm compared with the baseline methods?
>
> The term “without additional implementation cost” means that the selection event can be automatically computed when the network structure is changed. For example, we only need to use Keras AIP to add more layers or change the number of hidden nodes and the proposed method will automatically compute the selection event (e.g., we do not need to re-implement the selection event by ourselves). This is the reason why proposed method can be easily integrated with various CNNs.
>
>
> Regarding the complexity, we will provide the following paragraph in the revised paper:
>
> “In the literature of homotopy method (a.k.a. parametric programming), it is known that the actual computational cost differs significantly from the worst case. A well-known application of the homotopy method in the ML community is the Lasso regularization path, which also has the worst-case computational cost on the exponential order of the number of features, but the actual cost is known to be nearly linear order. This also applies to our proposed method.”
>
> The baseline methods are computationally cheaper than the proposed method because they sacrifice a significant amount of TPR. Our method maintains a significantly higher TPR than the baseline methods with a reasonable computational cost (i.e., linear order in practice).

---

### Official Review · Reviewer_R4SS · 2022-10-24

**Confidence:** 3
**Correctness:** 4
**Technical Novelty And Significance:** 2
**Empirical Novelty And Significance:** 2
**Recommendation:** 6

**Clarity, Quality, Novelty And Reproducibility:**

The paper is well written with high quality and clarity. The math derivation of SI for the salient region p-value calculation is also clear. The work is novel by applying SI for salient region detection. The code is also provided for reproducibility though I didn’t evaluate the code for reproducibility for the purpose of this review.

**Strength And Weaknesses:**

Strength:

1. The paper in general clearly explained the motivation of the work by identifying the problem with existing statistical inference framework, i.e. selection bias.

2. The authors employed the selective inference framework to address the issue of selection bias in the context of saliency region detection and reliability estimation.

3. The authors claimed that the proposed method is applicable to a wide class of CNNs, demonstrated by an analysis of piecewise linear network.

4. The authors claimed to have implemented the proposed method in an efficient way and provided the code for reproducibility.

5. The experiments provided also supported the claims made by the authors regarding the validity and applicability of the proposed method on practical problems.

Weaknesses:

1. Some parts of the math derivation are not very clear to me and may need further explanation. For example, between equations (6) and (7), “… we consider the following selection event…” for the global null test, the sign matching part is not very obvious to readers.

2. On the line after equation (12), it is claimed that the variable Z is normally distributed, which is an important assumption for subsequent derivations to calculate the p-value. It could be further explained why it is normally distributed.

3. The authors claimed that their Keras implementation of Auto-Conditioning is efficient, but provides no asymptotic analysis of the computational complexity of their algorithm. This is important when applying this method to practical applications.

4. When using the brain image dataset in the experiment, does it matter which image is selected as the reference for the proposed method to work? What is the effect of selecting different reference images?

5. The authors performed experiments using a synthetic dataset and a real brain image datasets, and compared the False Positive Rate and True Positive Rate of the proposed method and some other methods for the synthetic dataset. I suggest the authors do the same analysis for the real dataset, and do some more analysis with another natural image dataset, which is usually used to compare different saliency methods.


**Summary Of The Paper:**

In this paper, the authors proposed a method to quantify the reliability of the saliency region in the form of p-values by employing the selective inference (SI) framework. In the proposed method, the authors regard the salient region proposed by DNN explainability methods such as CAM as a selected hypothesis. Since the hypothesis proposed by the model is not independent of the data, instead of classical inference with selection bias, the authors proposed to use selective inference where the inference is conditioned on a selected event. The authors also demonstrated the validity of the method through numerical and real datasets.

The main contributions of this work:

1. The authors employed the selective inference framework to compute a p-value to quantify the reliability of a salient region proposed by a saliency model to overcome the selective bias problem.

2. The authors also claimed to have provided a Keras-based implementation to efficiently and flexibly conduct conditional SI for salient regions for a wide class of CNNs.

3. The authors conducted experiments on both synthetic and brain image datasets and claimed the proposed method is able to control the false positive rate (FPR) and has potential to be applied to practical applications.


**Summary Of The Review:**

This paper proposed to evaluate the reliability of a salient region with a p-value by employing the selective inference (SI) framework to address the selection bias problem with classical inference. The authors also claimed to have provided an efficient Keras-based implementation of the proposed method, and experiments have conducted to prove the validity of the proposed method. In my opinion, this work is well motivated, the solution is clearly explained, and the claims are also supported by empirical results. I suggest the authors to perform more experiments on natural images with additional analysis for deeper insights.

---

> ### Author Response · Authors · 2022-11-13
> **Our Responses to Reviewer R4SS**
>
> We thank the reviewer for your comments.
>
> > Some parts of the math derivation are not very clear to me and may need further explanation. For example, between equations (6) and (7), “… we consider the following selection event…” for the global null test, the sign matching part is not very obvious to readers.
>
> In the revised paper, we will provide the following paragraph to make clarification:
>
> “We would like to note that, in the case of global null test, we also considering the sign conditioning because the test statistic in global null test is defined as the sum of absolute difference which depends on the sign. In contrast, we do not consider the sign in the case of mean null test because the definition of test statistic only depends on the salient region.”
>
>
> > On the line after equation (12), it is claimed that the variable $Z$ is normally distributed, which is an important assumption for subsequent derivations to calculate the $p$-value. It could be further explained why it is normally distributed.
>
> In the revised paper, we will provide the following discussion:
>
> “As shown in proof of Lemma 1, $Z = \boldsymbol{\eta}^{\top} {\boldsymbol{X} \choose \boldsymbol{X}^{\rm ref}}$. Since $\boldsymbol{X}$ and $\boldsymbol{X}^{\rm ref}$ are corrupted with Gaussian noises as discussed in the beginning of Section 2, $Z$ follows normal distribution”
>
>
> > The authors claimed that their Keras implementation of Auto-Conditioning is efficient, but provides no asymptotic analysis of the computational complexity of their algorithm. This is important when applying this method to practical applications.
>
> Auto-Conditioning is a trick to ease implementation, not a way to reduce computational cost. Therefore, we did not provide the analysis of computational complexity. Auto-Conditioning has advantages in the same sense as automatic differentiation. Conventional selective inference (including Duy et al. (2022)) requires explicit coding of selection event, so even small changes to the network structure require reimplementation. Our auto-conditioning method takes advantage of the fact that selection events can be computed by forward propagation in each layer. Since selection events are implemented in the layer API of Keras, it is possible to automatically compute selection event. More specifically, the conditioning part of Equation. (9) can be automatically computed by simply adding/removing layers, i.e., just a few line modification is sufficient to conduct SI for different network structures (the reviewer can simply add/remove layers of the network in the 'demo.ipynb' file provided in the supplements and the code can be easily run). This auto-conditioning enables SI for hypotheses characterized at any layer in the network, which allows us to consider quantification of the statistical significance of saliency regions.
>
>
>
> > When using the brain image dataset in the experiment, does it matter which image is selected as the reference for the proposed method to work? What is the effect of selecting different reference images?
>
> Any reference image is fine as long as it is a null image (does not contain any true salient region) and has similar structure compared to the input image. However, as pointed out by reviewer TSgM, there might be situations where there is a misalignment between the input image and reference image and have not considered these cases in our method. A potential future improvement could be additionally performing a (matching) step to automatically find the appropriate region in the reference image before conducting a statistical test. If the matching operations can be represented as a set of linear inequalities, they can be easily incorporated to the proposed method.
>
>
> > The authors performed experiments using a synthetic dataset and a real brain image datasets and compared the FPR and TPR of the proposed method and some other methods for the synthetic dataset. I suggest the authors do the same analysis for the real dataset, and do some more analysis with another natural image dataset, which is usually used to compare different saliency methods.
>
> In this study, our main target application domain is biomedical fields in which the $p$-values are popularly used to quantify the statistical significance of the results discovered from medical images. On the other hand, the interpretation of the $p$-value is not always clear in the cases of natural images. As suggested by the reviewer, we conducted additional experiments for FPR and TPR comparisons in real brain image dataset. The results are shown in the following table:
>
>
> |       | Naive | Bonferroni | OC | Proposed |
> | ------ | ------ | ------ | ------ | ------ |
> | FPR | 0.72 | 0.00 | 0.06 | 0.07  |
> | TPR | N/A | 0.00 | 0.03 | 0.58 |
>
> Since the naive method fails to control the FPR, we did not consider its TPR. The Bonferroni is too conservative. The OC has low TPR due to the over-conditioning. The proposed method has the highest TPR while properly controlling the FPR.

---

> > ### Comment · Reviewer_R4SS · 2022-11-21
> > **Good improvement**
> >
> > I would like to thank the authors for updating the manuscript based on my feedback. I am satisfied with the author's explanation and glad to see the improved quality of the paper. I would like to keep the recommendation as "6: marginally above the acceptance threshold".

---

### Official Review · Reviewer_rBPS · 2022-11-03

**Confidence:** 2
**Correctness:** 3
**Technical Novelty And Significance:** 2
**Empirical Novelty And Significance:** 2
**Recommendation:** 5

**Clarity, Quality, Novelty And Reproducibility:**

Overall, the paper is clear and they also provided code for reproducibility. The code is commented and the quality and readable.

**Strength And Weaknesses:**


# A. Strengths
### A1. The topic of quantifying reliability for saliency maps is a topic of interest and relevant to the community.
### A2. The proposed approach is based on theoretical justification and overall the presentation is adequate.
### A3. Code and scripts to reproduce the paper are provided in the supplemental materials.


# B.  Weaknesses
### B1. Discussion regarding Duy et al (2022) seems vague. The paper states
> In this study, we docs on a more general problem setup.

In what ways? Empirical comparison in the same setup as Duy et al (2022)? The current description does not fully disclose this paper's contribution. Please clarify.


### B2. Justification for the test statistic in Eq. (4). The test statistic is defined as
> $
T(\mathbf{X}, \mathbf{X}^{\text{ref}}) = \sum_{i \in \mathcal{M}_{X}} |X_i - X_i^{\text{ref}}|
$

This is pixel-wise comparison of the elements within the salient region. However, pixel-wise differences are not particularly meaningful, especially, for capturing higher-level concepts in an image. Can the author clarify the motivation for using such test statistics?


### B3. What is the practical application for performing a two-sample test when the test-outcome depends on the provided reference? In particular, how is the reference image selected in practice, e.g., in real data experiments?


### B4. The paper states that for piecewise linear network
> This is also true for the saliency map function $\mathcal{A}_i(\mathbf{X})$.

I believe this statement is inaccurate, it depends on which saliency method is used.

Additionally, is this assumption true in practice? Typically, saliency methods are applied to classifier networks which contain softmax non-linearities for CNNs; similar issues with, the recently popular, transformer achirectures.

Hence, the title “Deep learning-driven” salient region seems to be a bit of an overstatement.


### B5. Some experimental details are not clear.
The network architecture is provided in the supplemental material, however, it is unclear how the network is trained and on what annotation.


### B6. Have the authors' considered other datasets beyond the brain image provided?

Evaluation of more general/large-scale computer vision datasets could strengthen the contribution of this paper. As the authors claimed contribution on
> develop a Keras-based framework for conducting the proposed selective inference ... without additional implementation cost.

It seems like the method can be easily applied to other datasets, why not demonstrate some more results?

### B7. Any thoughts on Bayesian approaches, e.g., [I],  to uncertainty? A discussion would be interesting to have.

[I] Kendall, Alex, and Yarin Gal. "What uncertainties do we need in bayesian deep learning for computer vision?." Advances in neural information processing systems 30 (2017).

### B8. The notation for $\tilde{\Sigma}$ should be clarified.


**Summary Of The Paper:**

This paper proposes a method for quantifying the reliability of saliency maps extracted from deep nets. Specifically, they perform a two-sample test, where the test statistic is based on the sum of differences between the given sample and a reference within salient regions. The main difficulty lies in how to compute the distribution of the test statistic, as the saliency depends on a deep net. To solve this challenge, they propose and prove an algorithm to compute this distribution for a family of piecewise linear networks. Empirical results are also provided on a synthetic dataset and a real-world brain image dataset.

**Summary Of The Review:**

I have some concerns about the usefulness of the developed statistical test and need some clarification regarding their model assumption. See the weaknesses section for more details.

---

> ### Author Response · Authors · 2022-11-13
> **Our Responses to Reviewer rBPS**
>
> We thank the reviewer for your feedback.
>
> > Discussion regarding Duy et al (2022) seems vague.
>
> We focus on a more general problem setup compared to Duy et al. (2022). More specifically, Duy et al. (2022) focus on the segmentation problem in which the hypotheses are characterized by the output layer whereas, in this paper, we generalize the approach so that hypotheses characterized by any internal nodes of the network can be considered, which enables us to quantify the statistical significance of saliency regions.
>
> Moreover, their method requires high implementation cost and is not flexible when the structure of a DNN is changed. As we will mention later, we develop a practical Keras-based conditional SI framework for a wide class of CNNs without additional implementation cost. In other words, we only need to modify the network structure by Kera’s APIs and the conditional SI framework will be automatically done by the proposed method.
>
> > Justification for the test statistic in Eq. (4).
>
> The reason why we defined the test statistic as the sum of absolute value for testing the global null test is because it is commonly used in the literature of multivariate two-sample test. In practice, the choice of the test statistic should depend on the objective of the task. Our method is applicable to the class of linear statistics and can be applied to a variety of other practical test statistics.
>
> > What is the practical application for performing a two-sample test when the test-outcome depends on the provided reference? In particular, how is the reference image selected in practice, e.g., in real data experiments?
>
> Our study is motivated by practical applications in medical image diagnosis. A medical doctor wants to confirm the correctness of the detected salient regions (e.g., abnormal regions) by comparing them with the corresponding region in an annotated reference image. Therefore, in biomedical practice, the reference image is provided by domain experts (e.g., medical doctors).
>
>
> > The paper states that for piecewise linear network
>
> We will update the sentence as follows:
>
> “We mainly focus on the saliency map for piecewise linear network. Therefore, we consider the saliency map function $\mathcal{A}_i(\boldsymbol{X})$ as a piecewise linear function. If there is any specific demand to use non-piecewise linear functions, we can apply a piecewise-linear approximation approach to these functions. We would like to note that a large class of deep networks can be represented (or approximated with sufficient accuracy) as a piecewise linear function (see, e.g., Bunnel et al. A unified view of piecewise linear neural network verification. NeurIPS 2018).”
>
>
> > Some experimental details are not clear.
>
> In the revised paper, we will additionally provide the information on how the network is trained. However, we would like to note that, since we are primarily interested in the reliability of a trained network given new inputs (i.e., post-hoc analysis), the validity of our proposed method does not depend on how the DNN is trained.
>
>
> > Have the authors' considered other datasets beyond the brain image provided?
>
> Currently, we are in the process of applying the proposed method to practical scientific problems, focusing on medical images such as CT images and pathological images, along with domain experts. We note that our main purpose of this submission to ICLR is to share the methodology and proof of the theory to machine learning community, and that the numerical experiments are merely to demonstrate that the correctness of the theory.
>
> > Any thoughts on Bayesian approaches? A discussion would be interesting to have.
>
> We haven’t pursued Bayesian direction yet, but we will discuss some Bayesian approaches (e.g., the paper that the reviewer suggested) in the revised paper. We would like to note that, in biomedical field which is the main target domain of this study, providing $p$-value obtained by frequentist approaches are essential for reproducibility of scientific finding.
>
> > The notation for $\tilde{\Sigma}$ should be clarified.
>
> We defined $\tilde{\Sigma}$ at the end of the Equation below Equation (7).

---

### Official Review · Reviewer_TSgM · 2022-11-04

**Confidence:** 3
**Correctness:** 4
**Technical Novelty And Significance:** 3
**Empirical Novelty And Significance:** 4
**Recommendation:** 6

**Clarity, Quality, Novelty And Reproducibility:**

Clarity and Quality:
- The quality of the writing is good. The codebase in the supplementary material is good.
- Just below equation (4), the authors mention eta_{M_x}, but this isn't defined anywhere nearby. What is this variable?
- The authors use unusual symbols for referencing figures. Clarity would be improved by using the hyperref or cleverref packages and typesetting references as "Figure 1", "Table 1", etc. I think this might actually be in the style guideline file.


Novelty:
The novelty is good overall. The technical contribution with auto-conditioning seems substantial.

A minor point: The authors say "To our knowledge, this is the first method that can provide valid p-values to statistically quantify the reliability of the DNN-driven salient regions", but Duy et al. (2022) (the closest work to this one) say pretty much the same thing in their introduction. The authors explain their contributions relative to Duy et al., but it might be worth qualifying this statement to reflect those contributions.

**Strength And Weaknesses:**

Strengths:
- The proposed auto-conditioning method seems quite novel (conceptually and technically)
- Interpretability is an important problem area
- Compared to baselines, the proposed method greatly reduces false positives.

Weaknesses:
- Since I am not very knowledgeable about selective inference, I'm not sure whether the proposed method actually allows one to "quantitatively evaluate the stability and reproducibility of DNN-driven salient regions". If the authors could include some commentary or clarification of how the proposed method actually improves the usability of saliency maps, that would make the paper much more approachable for the broader ICLR audience. For instance, is the practical impact that one doesn't need to worry about setting thresholds for detection (assuming knowledge of the noise distribution)?
- It seems like the proposed method might require knowing the noise distribution. Is this a limitation?
- The noise is pixel-wise, which doesn't seem to capture important forms of structural variation. For instance, the reference image and input image have slight misalignments, which could be a problem for the proposed method. The authors don't seem to discuss this.

**Summary Of The Paper:**

This paper proposes a conditional selective inference (SI) approach for obtaining p-values for saliency maps. By leveraging the piecewise linearity of ReLU networks, they develop a method for performing the necessary hypothesis testing in an automated fashion across arbitrary network architectures. In experiments, their method outperforms several baselines

**Summary Of The Review:**

Overall, this seems like a solid technical contribution in an area that people care about. I don't know much about the area, so I have low confidence, but I'm leaning towards acceptance after spending 2 hours reading the paper. I may be missing something important, though, and would be willing to adjust up or down depending on other reviews and author feedback.

--------------

Update after rebuttal:

The authors have addressed my concerns. I'm increasing my confidence in the accept decision. I'm not confident enough to increase the score to an 8 (the next highest increment).

---

> ### Author Response · Authors · 2022-11-13
> **Our Response to Reviewer TSgM**
>
> We thank the reviewer for your comments.
>
> > I'm not sure whether the proposed method actually allows one to "quantitatively evaluate the stability and reproducibility of DNN-driven salient regions".
>
> In our understanding, the stability and robustness of XAI is considered to be one of the important topics in machine learning community. To the best of our knowledge, the proposed method is the first to accurately quantify the statistical significance of saliency regions using $p$-value, a commonly used criterion in science and engineering, inspired by Duy et al. (2022). We believe that this study is an important step toward reliable explainable AI. In practice, the proposed method is directly useful in biomedical applications. Quantification of significance by $p$-value is essential for scientific findings in the biomedical field to ensure reproducibility, and this study provides a tool to quantify the reliability of AI-driven scientific findings. Besides, as suggested by the reviewer, another practical impact of the proposed method is that the false positive rate is guaranteed to be controlled for any detection threshold given by the analysts, i.e., the analysts do not need to worry too much about setting the threshold.
>
>
> > It seems like the proposed method might require knowing the noise distribution. Is this a limitation?
>
> Yes, this might be considered as a limitation. We would like to note that the requirement of normally distributed noise is fundamental, and it is used in almost all Selective Inference (SI) related studies that we cited. Moreover, we believe that it is not uncommon to assume Gaussian noise in computer vision and other signal processing problems.
>
> We understand that the robustness against violation from the normality assumption is desirable. Therefore, we conducted additional experiments to empirically check the performance of FPR control when the noise follows skew normal, Laplace, or t-distributions. The results are shown in the following tables. In general, the proposed method still maintains good performance in terms of FPR control at $\alpha = 0.05$.
>
>
> + FPR control in global null test
>
> |           |   n = 64    |   n = 256    |   n = 1024    |   n = 4096    |
> | ----------| ----------  | ----------  | ----------  | ----------  |
> | Laplace   | 0.025 | 0.035 | 0.045 | 0.033 |
> | Skew      | 0.043 | 0.053 | 0.057 | 0.054 |
> | t-dist    | 0.060 | 0.044 | 0.053 | 0.047 |
>
> + FPR control in mean null test
>
> |           |   n = 64    |   n = 256    |   n = 1024    |   n = 4096    |
> | ----------| ----------  | ----------  | ----------  | ----------  |
> | Laplace   | 0.060 | 0.050 | 0.060 | 0.048 |
> | Skew      | 0.050 | 0.055 | 0.043 | 0.044 |
> | t-dist    | 0.044 | 0.048 | 0.042 | 0.041 |
>
>
>
> > The noise is pixel-wise, which doesn't seem to capture important forms of structural variation. For instance, the reference image and input image have slight misalignments, which could be a problem for the proposed method. The authors don't seem to discuss this.
>
>
> Thank you for pointing out this point. We will add the following paragraph in Section 7 of the revised paper to discuss the current limitations and potential future improvement:
>
> “In current setting, we have not considered the situations where there is a misalignment between the input image and reference image. A potential future improvement could be additionally performing a (matching) step to automatically find the appropriate region in the reference image before conducting a statistical test. If the matching operations can be represented as a set of linear inequalities, they can be easily incorporated to the proposed method.”
>
> We conjecture that this is possible without difficulty because basic matching operations can be represented as linear selection event, i.e., by the set of linear inequalities. Then, we can simply add the selection event to the conditioning part of the conditional sampling distribution in Eq. (5).
>
>
> > Just below equation (4), the authors mention $\boldsymbol{\eta}_{\mathcal{M}_X}$, but this isn't defined anywhere nearby. What is this variable?
>
>
> The $\boldsymbol{\eta}_{\mathcal{M}_X}$ in Equation (4) is generally a vector in $\mathbb{R}^{2n}$ and it is defined depending on the test that we are interested in.
>
> Two examples of defining $\boldsymbol{\eta}_{\mathcal{M}_X}$ in the cases of global null test and mean null test are provided in the text just below Equation (4) and above Section 3.
>
>
> > A minor point: The authors say "To our knowledge, this is the first method that can provide valid p-values to statistically quantify the reliability of the DNN-driven salient regions"
>
> Thank you for suggesting the clarification. In the revised paper, we will update the sentence as follows:
>
> “We introduce valid $p$-values to statistically quantify the reliability of the DNN-driven salient regions inspired by Duy et al. (2022)”
>
> Please see also our answer to the $1^{\rm st}$ question by Reviewer rBPS below.

---

### Official Review · Reviewer_Duyc · 2022-11-07

**Confidence:** 2
**Correctness:** 3
**Technical Novelty And Significance:** 3
**Empirical Novelty And Significance:** 3
**Recommendation:** 6

**Clarity, Quality, Novelty And Reproducibility:**

This paper is well-written in my opinion. The proposed method is novel and provides new insights into saliency map detection.

The authors discuss the implementation in the paper and provide the code for review. I haven't verified if the experiments in this paper can be reproduced though.

**Strength And Weaknesses:**

Strengths:

1. This work looks technically sound.

2. The proposed method is correlated to statistical hypothesis tests. This idea is interesting and could be practically useful in some real-world scenarios, e.g., medical applications.

Weaknesses:

1. In this work, the saliency map is presumed to be generated by CAM (Zhou et al., 2016). We don't know how well the proposed method works with other CAM-like saliency map methods (e.g., Grad-Cam). Correspondingly, in the theory part, how the saliency map model A(\cdot) affects the proposed selective p-value is unclear.

2. How the proposed selective p-value corresponds to the correlation between the salient region and the reference region is unknown. In other words, can the selective p-value be aligned with the visual difference/similarity between salient regions and reference regions? One way to measure the linear correlation coefficient between a model saliency map and an empirical saliency map is to use correlation coefficient (CC). Please refer to \url{https://saliency.tuebingen.ai/results.html} for more details.

3. Last but not least, the experiments are less convincing as the proposed method is only evaluated on a synthetic dataset and a small-scale dataset. I cannot tell if the proposed method can work well on other general yet complicated real-world datasets.

**Summary Of The Paper:**

This work attempts to shed some light on the reliability of saliency maps.
Specifically, the authors propose a conditional Selective Inference (SI) method. The proposed method can provide valid p-values for statistically quantifying the reliability of the salient regions. The proposed method is evaluated on a synthetic dataset and a brain image dataset.

**Summary Of The Review:**

Overall, I think the proposed conditional Selective Inference method is interesting and insightful. However, my main concern lies in the empirical evidence. It would be better to see more experimental results on other datasets, e.g., a natural image dataset, like CIFAR.

---

> ### Author Response · Authors · 2022-11-13
> **Our Responses to Reviewer Duyc**
>
> We thank the reviewer for your feedback.
>
> > In this work, the saliency map is presumed to be generated by CAM (Zhou et al., 2016). We don't know how well the proposed method works with other CAM-like saliency map methods (e.g., Grad-Cam). Correspondingly, in the theory part, how the saliency map model $\mathcal{A}(\cdot)$ affects the proposed selective p-value is unclear.
>
>
> Our method is applicable as long as the saliency map function can be written as a piecewise linear function of an input vector. Since Grad-CAM satisfies this condition, application of the proposed method in the case of Grad-CAM is straightforward. Mathematically and theoretically, the proposed p-value is valid for any saliency map model. However, to make it practically useful, we mainly focus on a class of models that can be represented as piecewise linear functions, which enable us to tractably characterize the selection event.
>
>
> > How the proposed selective $p$-value corresponds to the correlation between the salient region and the reference region is unknown. In other words, can the selective p-value be aligned with the visual difference/similarity between salient regions and reference regions? One way to measure the linear correlation coefficient between a model saliency map and an empirical saliency map is to use correlation coefficient (CC).
>
>
> Providing selective $p$-value for correlation coefficient (CC) is interesting. However, our current method cannot be directly applied to that setting because the CC cannot be represented as a linear test-statistic, i.e., in the form of $\boldsymbol{\eta}^\top \boldsymbol{X}$. Extending the current framework to the case of non-linear test-statistic is one of our important future works.
>
>
>
> > Last but not least, the experiments are less convincing as the proposed method is only evaluated on a synthetic dataset and a small-scale dataset. I cannot tell if the proposed method can work well on other general yet complicated real-world datasets.
>
>
> We would like to note that the proposed method has strict theoretical guarantees in terms of Type I error control without any approximation. Therefore, it should work well for any datasets as long as the assumptions are satisfied regardless the size. Our intended applications are in fields where reliability assurance based on $p$-value is required, such as medical diagnosis and quality control. Currently, we are in the process of applying the proposed method to practical scientific problems, focusing on medical images such as CT images and pathological images, along with domain experts.

---

### Decision · Program_Chairs · 2023-01-20

**Decision:**

Accept: poster

**Justification For Why Not Higher Score:**

Not enough novelty, significance or likely impact to justify further promotion.

**Justification For Why Not Lower Score:**

This is the weakest paper among those I recommend for acceptance. I list the points in favor above. But I would not be upset if the paper was ultimately left out.

**Metareview: Summary, Strengths And Weaknesses:**

The paper proposes a method for quantifying statistical significance for saliency maps derived from deep models, in particular in the context of "explainability" evaluation. The focus of evaluation is on medical images. The work builds significantly upon prior work by Duy et al., but extended it in particular to allow for separate reference image.

Besides the relationship to the prior work, which I think the authors have resolved (leaving the question about the significance of the "delta"), the main concern is the limited nature of the test statistic, which effectively assumes pixel-wise alignment between the images. In my view that is indeed an issue, and a big limitation of the proposed method. There authors acknowledge this but are, in my opinion, a bit too cavalier about the likely ease of overcoming it.

Overall, after further discussion with the reviewers, I believe this paper is just barely above threshold. If it's accepted the authors can


**Note From Pc:**

if the above contains the word "oral" or "spotlight" please see: "oral" presentation means -> notable-top-5% and "spotlight" means -> notable-top-25%. As stated in our emails, we are disassociating presentation type from AC recommendations

**Summary Of Ac-Reviewer Meeting:**

The only reviewer I could get hold of was rBPS. They remain unconvinced by the justification for the aligned test. However, I have confirmed that rBPS doesn't object to accepting the paper. Thus, three of the reviewers are either in favor or neutral, which makes it possible in my view to accept. The main points in favor are the well defined task which is relevant, clear exposition, and to some degree, the fact that this paper is not about a minor modification in a "hot area" but something a bit different.